# The Cross-environment Hyperparameter Setting Benchmark for Reinforcement Learning

**Andrew Patterson, Samuel Neumann, Raksha Kumaraswamy, Martha White, Adam White**
Department of Computing Science, University of Alberta
{ap3,sfneuman,kumarasw,whitem,amw8}@ualberta.ca

## Abstract

This paper introduces a new benchmark, the Cross-environment Hyperparameter Setting Benchmark, that allows comparison of RL algorithms across environments using only a single hyperparameter setting, encouraging algorithmic development which is insensitive to hyperparameters. We demonstrate that the benchmark is robust to statistical noise and obtains qualitatively similar results across repeated applications, even when using a small number of samples. This robustness makes the benchmark computationally cheap to apply, allowing statistically sound insights at low cost. We provide two example instantiations of the CHS, on a set of six small control environments (SC-CHS) and on the entire DM Control suite of 28 environments (DMC-CHS). Finally, to demonstrate the applicability of the CHS to modern RL algorithms on challenging environments, we provide a novel empirical study of an open question in the continuous control literature. We show, with high confidence, that there is no meaningful difference in performance between Ornstein-Uhlenbeck noise and uncorrelated Gaussian noise for exploration with the DDPG algorithm on the DMC-CHS.

## 1 Introduction

One of the major benefits of the Atari suite is the focus on more general reinforcement learning agents. Numerous agents have been shown to exhibit learning across many games with a single architecture and a single set of hyperparameters. To a lesser extent, OpenAI Gym (Brockman et al., 2016) and DM control suite (Tassa et al., 2018) are used in the same way—though at times not all environments are used, raising the possibility of cherry-picking. As the ambitions of the community have grown, Atari and OpenAI Gym tasks have been combined into larger problem suites, with subsets of environments chosen to test algorithms. In many ways we are back to where we started with Cartpole, Mountain Car and the like: where environment-specific hyperparameter tuning and problem subselection is prominent. Instead of proposing a new and bigger challenge suite, we explore a challenging new benchmark and empirical methodology for comparing agents across a given set of environments, complementing the existing empirical toolkit for investigating the scalability of deep RL algorithms.

In order to make progress towards impactful applications of reinforcement learning and the broader goals of AGI, we need benchmarks that clearly highlight the generality and stability of learning algorithms. Empirical work in Atari, Mujoco, and simulated 3D worlds typically use networks with millions of parameters, dozens of GPUs, and up to billions of samples (Beattie et al., 2016; Espeholt et al., 2018). Many results are demonstrative, meaning that the primary interest is not the stability and sensitivity, nor what was required to achieve the result, rather that the result *could* be achieved. It is infeasible to combine these large scale experiments with hyperparameter studies and enough independent runs to support statistically significant comparisons. More evidence is emerging that such state of the art systems (1) rely on environment-specific design choices that are sensitive to minor changes to hyperparameters (Henderson et al., 2018; Engstrom et al., 2019), (2) are less data

Submitted to the 35th Conference on Neural Information Processing Systems (NeurIPS 2021) Track on Datasets and Benchmarks. Do not distribute.

efficient and stable compared with simple baselines (van Hasselt et al., 2019; Taïga et al., 2019), and (3) cannot solve simple toy tasks without extensive re-engineering (Obando-Ceron and Castro, 2021; Patterson et al., 2021). It is abundantly clear that modern RL methods can be adapted to a broader spectrum of challenging tasks—well beyond what was possible with linear methods and expert feature design. However, we must now progress to phase two of empirical deep RL research: focusing on generality and stability.

There is a growing movement to increase the standards of empirical work in RL. Noisy results, inconsistent evaluation practices, and divergent code bases have fueled calls for more open-sourcing of agent architecture code, experiment checklists, and doing more than three independent evaluations in our experiments (Henderson et al., 2018; Pineau et al., 2020). Digging deeper, recent work has highlighted our poor usage of basic statistics, including confidence intervals and hypothesis tests (Colas et al., 2018). Long before the advent of deep networks, researchers called out the environment overfitting that is rampant in RL and proposed sampling from parameterized variants of classic control domains to emphasize general methods (Whiteson et al., 2009). Finally, and most related to our work, Jordan et al. (2020) proposed a methodology to better characterize the performance of an algorithm across environments, evaluated with randomly sampled hyperparameters. We build on this direction, but focus on a simpler and more computationally frugal evaluation that examines the single best hyperparameter setting across environments, rather than a randomly sampled one, and allows for a smaller number of runs per environment.

Table 1: Chance of incorrect claims

|  | 3 runs | 10 | 30 | 100 |
|---|---|---|---|---|
| Acrobot | 47% | 31% | 22% | 1% |
| Cartpole | 7% | 0% | 0% | 0% |
| CliffWorld | 54% | 19% | 14% | 0% |
| LunarLander | 16% | 7% | 1% | 0% |
| MountainCar | 22% | 9% | 7% | 0% |
| PuddleWorld | 18% | 16% | 8% | 0% |

One reason we focus on computational efficiency is that computational limitations seems to be the primary culprit for misleading or incorrect claims in RL experiments. Experiments with many runs, many hyperparameters, and many environments can be computationally prohibitive. The typical trade-off is to use a smaller number of runs. Such a choice, however, can lead to incorrect conclusions. Table 1 shows the empirical probability of incorrectly ordering four reasonable RL algorithms across several domains often considered too small to draw meaningful conclusions. We ran each of the four algorithms 250 times on every domain and for every hyperparameter setting in an extensive sweep to get a high confidence approximation of the correct ordering between algorithms. We then used bootstrap sampling to simulate 10k papers—each using a small number of random seeds—and counted the frequency that incorrect algorithm orderings were reported. Even with 30 runs in these small domains, incorrect rankings were **not** uncommon. Further details are described in Section 5.

Another critical issue for algorithm evaluation is the difficulty in hyperparameters selection. Modern RL algorithms require tuning an increasing number of hyperparameters, greatly impacting the outcome of an experimental trial. As more hyperparameters are introduced, the computational burden of tuning grows exponentially. To combat this, several strategies have emerged in the literature including relying on default hyperparameter values (Schaul et al., 2016; Wang et al., 2016; Van Hasselt et al., 2016), tuning hyperparameters on a subset of domains (Bellemare et al., 2013), or eroding standards of sufficient statistical power for publication (Henderson et al., 2018; Colas et al., 2018).

Our new benchmark is designed to (1) standardize the selection of hyperparameters, (2) evaluate stability over runs, (3) be computationally cheap to run, and (4) be easy to use. We propose the Cross-environment Hyperparameter Setting Benchmark (CHS). The basic idea is simple: an algorithm is evaluated on a set of environments, using the best hyperparameter setting across those environments, rather than per-environment. Though conceptually simple, this methodology is not widely used. We first address some of the nuances in the CHS, namely how to standardize performance across environments to allow for aggregation, how to allow for robust measures of performance, and finally how to reduce computation to make it more feasible to use the CHS. We evaluate the effectiveness of the CHS itself by examining the stability of the conclusions from the CHS under different numbers of runs. We then demonstrate that the CHS can result in different conclusions about algorithms compared to the conventional *per-environment tuning* approach and the more recent approach of using a subset of environments for tuning. We conclude with a larger demonstration of the CHS on DM Control Suite.

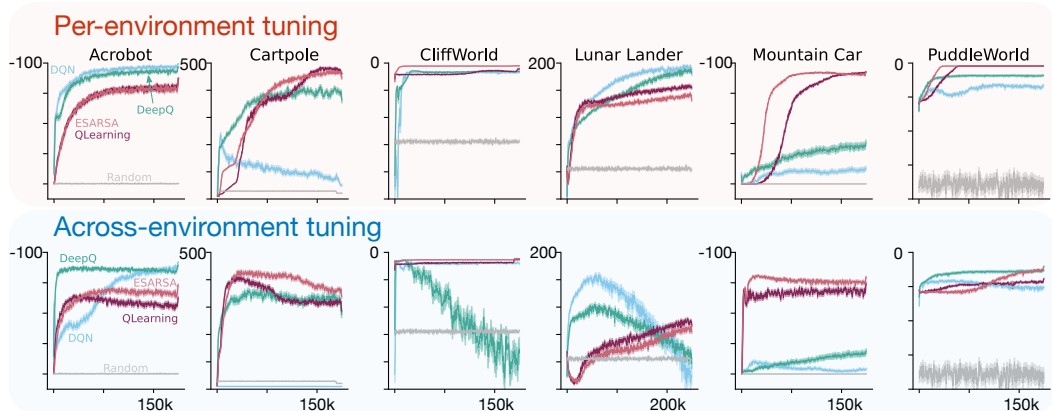

Figure 1: An example experiment comparing four algorithms across six different environments. Each learning curve shows the mean and standard error of 250 independent runs for each algorithm and environment. Hyperparameters are selected using three runs of every algorithm, environment, and hyperparameter setting. **Top** shows the learning curves when the best hyperparameters are chosen for each environment individually. **Bottom** shows the learning curves when hyperparameters are chosen according to our benchmark, the CHS.

## 2 Contrasting Across-Environment versus Per-Environment Tuning

In this section, we introduce the basic procedure for the CHS and provide an experiment showing how it can significantly change empirical outcomes compared to the conventional per-environment tuning approach. We provide specific details for each step later and here focus on outlining the basic idea and its utility.

The CHS consists of the following four steps summarized in the inset figure below. We assume we are given a set of environments and a set of hyperparameters for the algorithm we are evaluating.

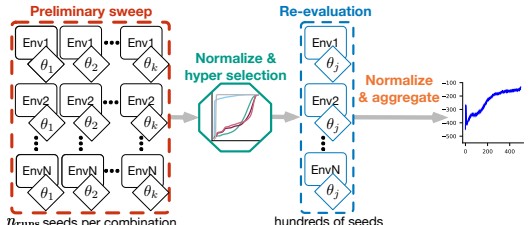

**Step 1 (Preliminary Sweep)** Run the algorithm for all hyperparameters and all environments, for $n_{\text{runs}}$ runs (i.e., $n_{\text{runs}} < 30$) and record the performance of every combination. The performance could be online average return per step.

**Step 2 (Normalization)** Normalize the scores across environments to be in $[0, 1]$. We use CDF normalization, which is described in Section 4.

**Step 3 (Hyperparameter Selection)** Select the hyperparameter setting with the highest score averaged across environments.

**Step 4 (Re-evaluation)** With the single best hyperparameter setting, use many more runs in each environment (e.g. 100) to produce a more accurate estimate of performance.

The last step is more lightweight than it appears since only a single hyperparameter setting is used for all environments. Executing 100 or more runs for every hyperparameter setting would likely be prohibitive. The trick is to use a small $n_{\text{runs}}$ in the Preliminary Sweep, saving compute, and a larger number of runs in the Re-evaluation step. This contrasts the conventional *per-environment tuning* approach of choosing hyperparameter settings which maximize performance on each environment individually, which can be more sensitive when $n_{\text{runs}}$ is small.

We now show an experiment comparing the CHS and this conventional approach in Figure 1. The per-environment tuning approach highlights the ideal behavior of an algorithm per environment, whereas the CHS highlights the (in)sensitivity of an algorithm across environments. Experimental details can be found in Section 5. The environments are relatively simple (most coming from the classic control suite of OpenAI Gym (Brockman et al., 2016)) but difficult enough for our purposes: no one algorithm could reach near optimal performance in all environments.

The CHS does not rank the algorithms differently than with per-environment tuning, but CHS does alert us to potential catastrophic failure of some algorithms. The neural network DeepQ agent performs terribly in Cliffworld and Lunar Lander under CHS, but appears reliable under the per-environment approach. What is going on? Forced to select only one hyperparameter

3

128 across environments, the best outcome is to sacrifice performance in Cliffworld and Lunar Lander—
129 achieving worse performance than a uniform random policy.

## 3   Performance Distributions

131 In this section, we describe the distribution and random variables underlying an RL experiment. This
132 formalism allows us to reason about the summary statistics we consider for the CHS in the next
133 section. We also visualize these distributions to provide intuition on the properties of the summary
134 statistics of these distributions and the implications for the single performance numbers used in RL.

135 In an RL experiment, we seek to describe the performance distribution of an algorithm for each
136 hyperparameter setting $\theta \in \Theta$, denoted as $\mathbb{P}(G, E \mid \theta)$ where $G$ is a random variable indicating
137 the performance of an algorithm on a given environment, $E \in \mathcal{E}$. Most commonly, we report
138 an estimate of the average performance conditioned on environment and hyperparameter setting,
139 $g(E, \theta) \approx \mathbb{E}[G \mid E, \theta]$ using a sample average and some measure of uncertainty about how accurately
140 $g(E, \theta)$ approximates $\mathbb{E}[G \mid E, \theta]$.

141 The environment can be seen as a random variable for many RL experiments. The most common
142 case is to specify a set of MDPs that the authors believe represent the important applications of their
143 new algorithm. If results are uniformly aggregated across these environments, then this corresponds
144 to assuming a uniform distribution over this set of environments. Other times, random subsets of
145 environments from environment suites are chosen; the performance estimate on this subset provides
146 an estimate of performance across the entire suite. The idea of evaluating algorithms over a random
147 sample of MDPs has been studied explicitly previously. For example, the parameters determining
148 the physics of classical control domains were randomized and sampled to avoid domain overfitting
149 (Whiteson et al., 2009), and randomly generated MDPs (Archibald et al., 1995) have been used to
150 evaluate new algorithmic ideas (Seijen and Sutton, 2014; Mahmood et al., 2014; White and White,
151 2016). If we subselect after running the algorithms, then we bias the distribution over environments
152 towards those with higher performance.

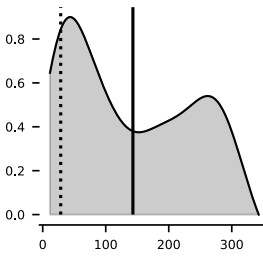

Figure 2:   Performance
distribution   $\mathbb{P}(G \mid E, \theta)$
on Cartpole with hyper-
parameter stepsize=$2^{-9}$.

Let us look at an example of these performance distributions to gain some
intuition for estimating statistics like the expected performance. Con-
sider the action-value nonlinear control method DQN, using the Adam
optimizer (Mnih et al., 2013; Kingma and Ba, 2015), on Cartpole (Barto
et al., 1983). We fix the hyperparameter setting $\theta$ to the default values
from Raffin et al. (2019). For this fixed environment, all randomness
is due to sampling algorithm performance on this environment, namely
sampling $G$ according to $\mathbb{P}(G \mid E, \theta)$. The performance, $G$, is the average
episodic return over all episodes completed during 100k learning steps.
This environment is considered solved for $G > 400$. We repeat this pro-
cedure for 250 independent trials to estimate the distribution $\mathbb{P}(G \mid E, \theta)$,
shown in Figure 2, with x-axis possible outcomes of $G$ and y-axis the
probability density. The vertical solid line denotes mean performance,
and the vertical dotted line denotes mean performance of a random policy.

167 Figure 2 is a typical example of the performance of an RL algorithm over multiple independent
168 trials. In this case, DQN is more likely to fail than to learn a policy which solves this relatively
169 simple environment. It is common practice to run an RL algorithm for some number of random
170 seeds—effectively drawing samples of performance from this distribution—then reporting the mean
171 over those samples (solid vertical line).

172 There are two implications from observing this bimodal performance distribution. First, using the
173 expected value of this distribution as the summary statistic does not aptly demonstrate that the poor
174 performance of DQN on Cartpole is due to occasional catastrophic failure—performing worse than
175 or equivalent to a random policy. Instead, mean performance might lead us to wrongly conclude that
176 DQN on Cartpole usually finds a sub-optimal, yet better than random, policy. An alternative might
177 be to consider percentile statistics or, if the goal is to evaluate mean performance, to avoid drawing
178 strong conclusions about individual runs.

179 If the goal is to report mean performance, then a second issue arises. Estimating the mean of these
180 non-normal performance distributions can be challenging. In Figure 2, approximately 70% of the

density is around a mode centered at 20 return, and the remaining 30% is around a mode centered at 250 return. As a result, sample means constructed with only three runs are varied and skewed.

Further, to report the average performance of the best performing hyperparameter—that is $\max_{\theta \in \Theta} \mathbb{E}[G \mid \theta, E]$—we must first reliably estimate the conditional expected performance for each hyperparameter. Computing this expectation can require a large number of samples to obtain a reasonable estimate for each hyperparameter. This results in a tradeoff between measuring sensitivity and stability: between the breadth of hyperparameter settings that can be studied and the accuracy to with which we can feasibly evaluate each hyperparameter.

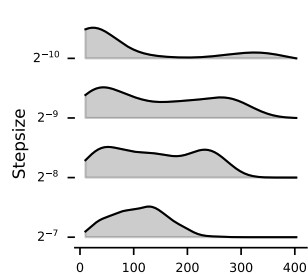

The summary statistic used to select hyperparameters also interacts with the form of the performance distribution. In the inset figure on the **left** we show the performance distribution across four different choices of stepsize parameter of DQN in **Cartpole**. If we are interested only in the highest best case performance, then $2^{-10}$ is preferred. However, if we are particularly concerned with reducing the chances of catastrophic failure (i.e., highest worst case performance), then a stepsize $2^{-7}$ is preferred. The most common case is to report results for the stepsize with the highest average performance. In this case, a stepsize of $2^{-9}$ would be preferred.

These performance distributions can also look quite different for different environments, even with the same algorithm. For Cartpole (above), the distribution is increasingly long-tailed with smaller stepsizes. For **Puddle World**, shown in the inset figure on the **right**, the distributions are always bimodal with one mode around -600 return and a second mode around -200 return. With smaller stepsizes, the density around the better performance mode increases, shifting the mean of the distribution. Peak performance does not change; rather the probability that DQN has a good run is higher with small stepsizes. This analysis of performance distributions raises an important question: do current RL algorithms have consistent hyperparameter settings which perform well across many environments?

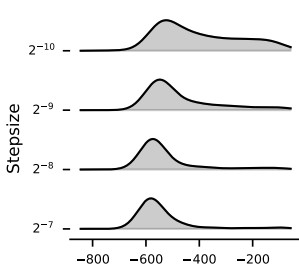

# 4 The Cross-environment Hyperparameter Setting Benchmark

In this section, we describe our new benchmark for evaluating RL algorithm across environments, the Cross-environment Hyperparameter Setting Benchmark (CHS). Although it seems natural to evaluate across environments, standard empirical practice in RL is not done this way. Understanding across-environment sensitivity aligns nicely with the intent of sensitivity analysis: elucidating how well an algorithm might perform on new environments without extensive hyperparameter tuning. We argue that the CHS 1) better aligns empirical practice with the goals of applied RL, 2) is computationally feasible even in complex environments, 3) provides novel insights on old ideas (even with small environments), and 4) reduces the chances of accidentally publishing incorrect conclusions due to statistical noise.

We now reiterate the procedure for the CHS with more details than the high-level procedure given in Section 2. The first step (**preliminary sweep**) is to draw a small number of samples $n_{\text{runs}}$ from $\mathbb{P}(G \mid \theta, E)$ for every hyperparameter setting and environment and get the summary estimate $g(E, \theta)$ from those samples. Typically, we compute $g(E, \theta)$ as a sample average to estimate $\mathbb{E}[N_E(G) \mid E, \theta]$, where $N_E : \mathbb{R} \to \mathbb{R}$ is a **normalization** function that we describe below. Then we aggregate across environments to estimate $g(\theta) \approx \mathbb{E}[\mathbb{E}[N_E(G) \mid E, \theta]]$, where the outer expectation is with respect to environments. Then we **select** a single hyperparameter setting with $\theta_{\text{CHS}} = \arg\max_{\theta \in \Theta} g(\theta)$. Finally, we draw a large number of samples from $\mathbb{P}(G \mid \theta_{\text{CHS}}, E)$ for every environment and report the same summary statistics $g(E, \theta_{\text{CHS}})$ and $g(\theta_{\text{CHS}})$ (**re-evaluation**).

In order to compute the expectation over environments we must normalize the performance measures. Generally, we cannot expect each environment to produce normalized performance numbers. A comprehensive discussion of normalization methods is given in Jordan et al. (2020). We use a lightly modified version of the CDF normalization method from Jordan et al. (2020), $N_E(G) = \text{CDF}(G, E)$, which is itself an instance of probabilistic performance profiles (Barreto et al., 2010).

We first collect the performance of each algorithm, environment, and hyperparameter tuple. Then, let $\mathcal{P}_E$ be the pool of performance statistics $g$ for every agent—namely a run of each algorithm and hyperparameter pair—for a given environment $E$. Our goal is to take a given agent's performance $x \in \mathcal{P}_E$ and return a normalized performance. The CDF normalization, for this $x$ in this environment, is

$$\text{CDF}(x, E) = \frac{1}{|\mathcal{P}_E|} \sum_{g \in \mathcal{P}_E} \mathbf{1}(g < x)$$

where $\mathbf{1}$ is the indicator function. This mapping says: what percentage of performance values, across all runs for all algorithms and all hyperparameter settings, is lower than my performance $x$ on this particular environment $E$? For example, if $\text{CDF}(x, E) = 0.25$, then this agent's performance is quite low in this environment, as only 25% of other agents' performance was worse and 75% was higher, across agents tested. This normalization accounts for the difficulty of the problem, and reflects relative performance amongst agents tested. Note that this normalization uses an empirical CDF, rather than the true CDF for the environment and set of hyperparameters and agents. This means there is a small amount of bias when estimating $\mathbb{E}[\mathbb{E}[N_E(G) \mid E, \theta]]$. This bias dissipates with an increasing numbers of samples and equally impacts all compared algorithms.

Selecting hyperparameters with the CHS can require significantly fewer samples compared with conventional per-environment tuning. Per-environment tuning requires a sufficiently accurate estimate of the conditional expectation $\mathbb{E}[G \mid E, \theta]$ for every $\theta \in \Theta$ and for every $E \in \mathcal{E}$, requiring a number of runs proportional to $|\Theta||\mathcal{E}|$. The CHS, on the other hand, requires only an accurate estimate of $\mathbb{E}[N_E(G) \mid \theta] = \mathbb{E}[\mathbb{E}[N_E(G) \mid E, \theta]]$ which requires a number of runs proportional only to $|\mathcal{E}|$. By designing a process which selects hyperparameters first using a smaller number of runs, we can reserve more computational resources for re-evaluation. Once we select the best hyperparameters, the cost of collecting samples is independent of $\Theta$, and so we can decouple the precision of our performance estimate from the number of hyperparameter settings that we evaluate for each algorithm.

Finally, we can contrast this benchmark with a recent evaluation scheme that uses random hyperparameter selection (Jordan et al., 2020). In order to capture variation in performance due to hyperparameter sensitivity, Jordan et al. (2020) treats hyperparameters as random variables and samples according to an experimenter-designated distribution over hyperparameters, reporting the mean and uncertainty with respect to this added variance, similar to the procedure used in Jaderberg et al. (2016). This evaluation methodology provides some insight into the difficulty of tuning, though requires a sensible distribution over hyperparameters to be chosen. The CHS, on the other hand, asks: is there a hyperparameter setting for which this algorithm can perform well across environments? It motivates instead identifying that single hyperparameter, and potentially fixing it in the algorithm, or suggesting that the algorithm needs to be improved so that such a hyperparameter could feasibly be found. Both of these strategies help identify algorithms that are difficult to tune, but the CHS is easier to use and computationally cheaper.

## 5   Evaluating the Cross-environment Hyperparameter Setting Benchmark

In this section, we evaluate the CHS by comparing four algorithms across several classic control environments. For this evaluation, we require environments where hundreds of independent samples of performance can be drawn across a large hyperparameter sweep in a computationally tractable way. We emphasize that this is not a general requirement of the CHS and is required only in this case of evaluating the CHS's responsiveness to perturbations in the experimental process. Because these classic control environments are cheap to run and provide meaningful insights in differentiating modern RL algorithms (Obando-Ceron and Castro, 2021), we name this specific benchmark the Small Control CHS (SC-CHS). In Section 6 we provide a realistic demonstration of the CHS on a larger dataset with a more complex algorithm.[1]

For the following investigations, we compare two deep RL algorithms based on DQN (Mnih et al., 2013) and two control algorithms based on linear function approximation using tile-coded features (Sutton and Barto, 2018). The deep RL algorithms, DQN and DeepQ, differ only in their loss: DQN uses a clipped loss and DeepQ uses a mean squared error. For the two tile-coding agents, QLearning is off-policy and bootstraps using the greedy action, while ESARSA is on-policy and bootstraps using an expectation over actions. Further details on the algorithms can be found in Appendix C.

---

[1]All code can be found at `https://github.com/andnp/single-hyperparameter-benchmark`.

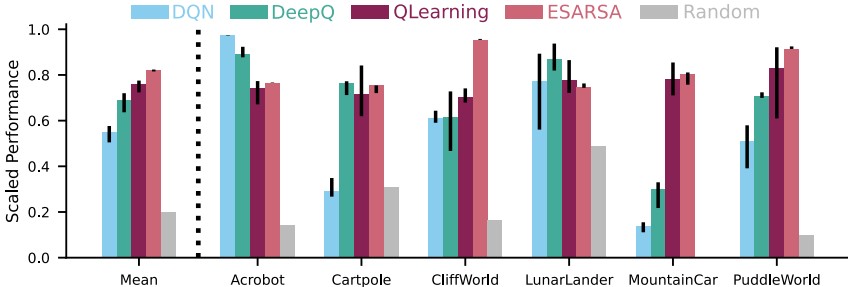

Figure 3: Applying the CHS to 10k simulated experiments. Error bars show 95% bootstrap confidence intervals. Although only three runs were used to select hyperparameters, conclusions about algorithm ranking using the CHS are perfectly consistent across all 10k experiments.

The SC-CHS consists of a suite of classic control environments commonly used in RL: Acrobot (Sutton, 1996), Cartpole (Barto et al., 1983; Brockman et al., 2016), Cliff World (Sutton and Barto, 2018), Lunar Lander (Brockman et al., 2016), Mountain Car (Moore, 1990; Sutton, 1996), and Puddle World (Sutton, 1996). We used a discount factor of $\gamma = 0.99$ and a maximum episode length of 500 steps (except in Cliff World which had a maximum length of 50 steps). We ran all algorithms for 200k learning steps on each environment except Lunar Lander, where we used 250k learning steps to ensure all algorithms have reliably converged. Further details motivating this choice of environments can be found in Appendix C.1.

We swept over several hyperparameter settings. For all algorithms we swept eight stepsize values, $\alpha \in \{2^{-12}, 2^{-11}, \ldots, 2^{-5}\}$ for the deep RL algorithms and $\alpha \in \{2^{-9}, 2^{-8}, \ldots, 2^{-2}\}$ for the tile-coded algorithms. The deep RL algorithms used experience replay and target networks, so we swept over replay buffer sizes of $\{2000, 4000\}$ and target network refresh rates of $\{1, 8, 32\}$ steps where a one step refresh indicates target networks are not used. The algorithms with tile-coding learn online from the most recent sample; we select number of tiles in each tiling in $\{2, 4, 8\}$ and number of tilings in $\{8, 16, 32\}$. More details on the other hyperparameters and design decisions are in Appendix C.

**Variance over simulated experiments.** Here we demonstrate that the CHS provides low variance conclusions over 10k simulated experiments using the benchmark. We use bootstrap sampling to compute 10k sample means over three random seeds for every algorithm, environment, and hyperparameter to first select hyperparameters using the CHS. We then evaluate the performance of each algorithm on each environment with 250 independent runs for the selected hyperparameter settings and compare the conclusions for each of the 10k simulated experiments.

Figure 3 demonstrates the consistency of conclusions made using the CHS across 10k simulated experiments. Using the CHS we would rank algorithms from best to worst ESARSA, QLearning, DeepQ, and DQN on this benchmark, and this ranking was successfully detected in every experiment. Conclusions on individual environments are less consistent. This is because selecting one hyperparameter across all these environments was difficult. In some runs, performance in one environment was sacrificed for the performance in the others; in another run, it was a different environment.

We provide more insight into the difficulty of selecting a single hyperparameter across problems, in Appendix B.1. We additionally show that the distribution of selected hyperparameters with the CHS is narrow and consistent over simulated experiments, unlike parameters chosen independently for each environment. Because conclusions are often drawn by aggregating results over environments—either formally as in the CHS or informally by counting the number of environments where an algorithm outperforms others—reporting results over a consistent and narrow distribution of hyperparameters leads towards lower variance claims and greater reproducibility. We include results selecting hyperparameters according to the worst-case performance across environments in Appendix B.4; the results are highly similar, albeit slightly lower variance.

The cost of running a single experiment represented in Figure 3 is quite low. The deep RL algorithms test 48 hyperparameter settings at a cost of 20 minutes per run, while the tile-coded algorithms test 72 settings at the cost of two minutes per run. Timings are with respect to a modern 2.1Ghz Intel Xeon processor. This comes out to a total of 1762 hours of CPU time to complete three runs for hyperparameter selection and 250 runs for evaluation, cheaper than the experiment using 10 runs and conventional per-environment tuning shown in Table 1 which cost approximately 2208 hours. The

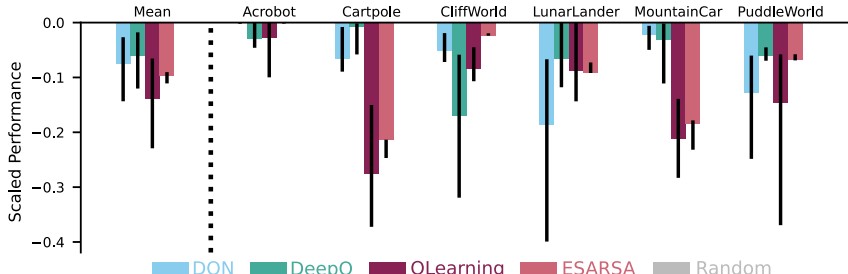

Figure 4: The change in performance for each algorithm on every environment when using the CHS versus conventional per-environment tuning. A larger drop in performance indicates a larger degree of environment overfitting when results are reported with per-environment tuning. Error bars show 95% confidence intervals over 10k bootstrap samples.

CHS successfully detected the correct ordering of algorithms in every trial, while the conventional per-environment tuning experiment failed to detected the correct ordering with surprising frequency.

**The CHS is a less optimistic measure of performance.** A motivating factor for the CHS is providing a more challenging benchmark to test across-environment insensitivity to selection of hyperparameters. Because algorithms are limited to selecting a single champion hyperparameter setting—as opposed to selecting a new hyperparameter setting for every environment—we expect a considerable drop in performance under the CHS. We evaluate the extent of this performance drop for our four algorithms by first computing near optimal parameters $\theta^* \in \Theta$ for each environment using the full 250 random seeds to obtain high confidence estimates of average performance $\mathbb{E}[N_E(G) \mid E, \theta^*]$. We then apply the CHS to select hyperparameters for each algorithm using three random seeds for 10k simulated experiments. We report sample estimates of $\mathbb{E}[N_E(G) \mid E, \theta^*] - \mathbb{E}[N_E(G) \mid E, \theta_{\text{CHS}}]$.

In Figure 4 we can see there is substantial drop in reported performance when using the CHS versus per-environment tuning. The variance is high, indicating that for some runs, the performance drop was substantial: almost 0.4 under our normalization between [0,1]. Algorithms with a large drop in performance indicate more environment-specific overfitting under per-environment tuning. Because we swept over many more hyperparameter settings for the tile-coding algorithms than for the deep RL algorithms—72 settings versus 48 settings—it is unsurprising that per-environment tuning led to far more environment overfitting in the tile-coding algorithms.

**Tuning on a subset of environments.** An empirical practice that is highly related to the CHS is using a subset of environments to select hyperparameters, then reporting the performance of the selected hyperparameters across an entire suite of environments. We refer to this practice as *subset-CHS*. This practice is used in the Atari suite for example, where it was suggested to use five of the 57 games for hyperparameter tuning (Bellemare et al., 2013). To investigate the variance of conclusions using the subset-CHS, we run 10k simulated experiments using two of our six environments to select hyperparameters. For each of the simulated experiments, we randomly select two environments to use for hyperparameter selection. To reduce the variance, we allow each algorithm 100 runs of every hyperparameter setting on every environment to perform hyperparameter selection, then evaluate the performance on the full 250 runs for the hyperparameter selected by the subset-CHS. More results, including with varying number of runs and environments used for hyperparameter selection, can be found in Appendix B.

In Figure 5, we see that the ordering of algorithms is extremely high-variance—especially compared to Figure 3 which uses all six environments to select hyperparameters and only three runs. This result also illustrates large differences between individual environments, where the variance on Lunar Lander—especially for DQN—suggests that hyperparameters selected for other environments are likely to cause worse-than-random performance on Lunar Lander. At least among the four tested algorithms, it is clear that hyperparameter sensitivity is too high to use environment subselection to reduce the computational burden of hyperparameter tuning.

**Bias of the CHS.** Both the CHS and conventional per-environment tuning use biased sample estimates due to the maximization over hyperparameters. The bias due to maximization over random samples is exaggerated both as the set $\Theta$ grows and as the number of samples used to evaluate $\mathbb{E}[G \mid E, \theta]$ shrinks. We first estimate the true per-environment maximizing parameters $\theta^*$ and the true CHS parameter $\theta_{\text{CHS}}^*$ using 250 samples for every hyperparameter setting and environment.

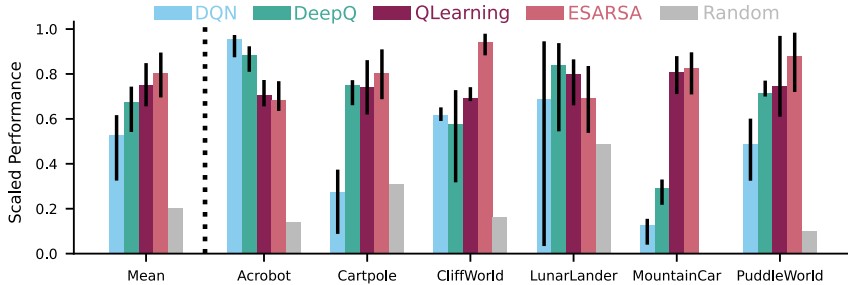

Figure 5: Performance of each algorithm over 10k bootstrap samples, where sample means are computed with 100 runs. Each bootstrap sample randomly selects two environments for hyperparameter tuning, then evaluates the chosen hyperparameter setting on all six environments with 250 runs. Error bars show 95% bootstrap confidence intervals.

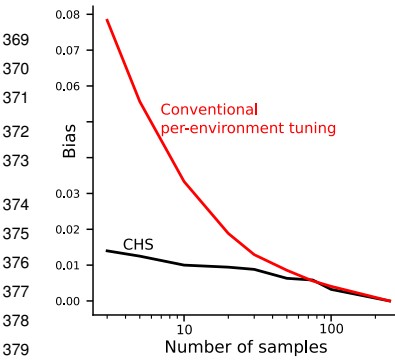

Figure 6: Bias of the CHS vs. per-environment tuning.

We then resample three samples per hyperparameter and environment to simulate an experiment using three seeds to compute sample averages, we select the maximizing parameter of these sample averages, $\hat{\theta}$, and we report $\mathbb{E}[G \mid E, \theta^*] - \mathbb{E}[G \mid E, \hat{\theta}]$. The corresponding procedure is used for the CHS.

In Figure 6, we report the bias of each procedure applied to DQN and the small control domain suite. On the vertical axis we report the bias and on the horizontal axis we show the number of random seeds used to select hyperparameters. As both procedures approach a sufficiently large number of samples to select hyperparameters, the bias of these procedures approaches zero. However when using few random seeds—for instance ten or fewer as is common in the literature—the bias of the conventional method is several times larger than that of the CHS. As a result of this overestimation bias, it is common for results in the literature to present highly optimistic results especially for algorithms with more hyperparameters.

## 6 A Demonstrative Example of Using the CHS

We finish with a large-scale demonstration of our benchmark across the 28 environments of the DMControl suite (Tassa et al., 2018), which we will call the DMC-CHS. For this comparison, we test an open hypothesis in the continuous control literature: does Ornstein-Uhlenbeck (OU) noise (Uhlenbeck and Ornstein, 1930) improve exploration over naive uncorrelated Gaussian noise? Autocorrelated noise for exploration was shown to be beneficial for robotics (Wawrzyński, 2015), inspiring the use of an OU noise process for DDPG (Lillicrap et al., 2016), where a single set of hyperparameters was used across 20 Mujoco environments using five seeds. Later work replaced OU noise with Gaussian noise, noting no difference in performance (Fujimoto et al., 2018; Barth-Maron et al., 2018), but without empirical support for the claim. To the best of our knowledge, no careful empirical investigation of this hypothesis has yet been published.

To apply the DMC-CHS, we first evaluate 36 hyperparameter settings with three runs per environment, for a total of 84 runs to estimate $\mathbb{E}[N_E(G) \mid \theta]$ for each $\theta \in \Theta$. Then we use 30 runs to evaluate the chosen $\theta_{\text{CHS}}$ for a total of 840 runs to estimate $\mathbb{E}[N_E(G) \mid \theta_{\text{CHS}}]$. We report the swept hyperparameters as well as the selected $\theta_{\text{CHS}}$ in Appendix B.5. We use 1k bootstrap samples to compute confidence intervals and report the overall findings in the table in Figure 7. We find that OU noise does not outperform Gaussian noise on the DMC-CHS. Considering even the extremes of the confidence intervals there is no meaningful difference in performance between these exploration methods, suggesting further runs would be unlikely to change our conclusion. We visualize the performance of OU noise on the complete suite, considering Gaussian noise experiments as a baseline in Figure 7. This visualization summarizes whether, and to what degree, OU noise improves upon Gaussian noise in each environment of the DMControl suite. In only 10 of the 28 environments, OU noise improves upon Gaussian noise, with a large improvement only in the *WalkerRun* environment. Additional results are included in Appendix B.5.

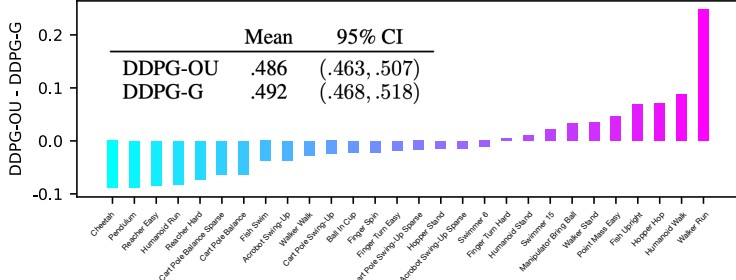

Figure 7: Comparing DDPG using OU noise vs. Gaussian noise across the DMControl suite. The inset table shows the mean performance with 95% confidence interval for the two versions of DDPG used in these experiments. Visualized in the bar plot is the performance of DDPG with OU noise, per environment in the suite, considering DDPG with Gaussian noise as a baseline.

## 7  Conclusion

In this work, we introduced a new benchmark for evaluating RL algorithms across environments, but perhaps more important are the insights we gained. Of the five algorithms we tested (including DQN and DDPG), none exhibited good performance on our CHS benchmark; aligning with the common view that we do not yet have generally applicable RL algorithms. The CHS benchmark produces reliable conclusions with only three runs in the preliminary sweep while providing a new challenging aspect to small computationally-cheap environments, allowing small university labs and tech giants alike to conduct rigorous and meaningful comparisons. Finally, prior work has disagreed on the benefit of using OU or Gaussian noise in DDPG on Mujoco-based environments. Perhaps some combination of too few runs, using default hyperparameters, or problematic environment sub-selection yielded conflicting results. Our results with CHS suggest there is no significant performance difference across a suite of 28 Mujoco environments, putting this debate to bed. The CHS benchmark can play a role uncovering falsehoods and resolving disputes.

The CHS is a general procedure for evaluating performance across environments. We provide two example instantiations of the CHS, the SC-CHS for discrete action control on small domains and the DMC-CHS for continuous control on large simulated environments, however the CHS can also be extended to use arbitrary environment sets to allow targeted evaluation across environments with certain desireable properties. For example, the taxonomies of Atari games identified in Bellemare et al. (2016), the off-policy evaluation environments used in Sutton et al. (2009), or the taxonomy of exploration environments from Yasui et al. (2019) are each sets of environments that have been previously identified and used across the literature. Applying the CHS to any one of the environment sets provides a new challenge, and in some small way can push us towards generally applicable RL agents.

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
