# OpenReview forum: "The Cross-environment Hyperparameter Setting Benchmark for Reinforcement Learning"
_NeurIPS.cc/2021/Track/Datasets_and_Benchmarks/Round1 — Submitted to NeurIPS 2021 Datasets and Benchmarks Track (Round 1)_

### Official Review · Reviewer_DfMW · 2021-07-01
**RL benchmark focussing on stability and generality**

**Rating:** 8
**Confidence:** 3

**Strengths:**

Performance of SHB appears to heavily depend on the inclusion / exclusion of single environments. For example, the authors demonstrate how under SHB, the algorithm DeepQ sacrifices performance in Cliffworld and Lunar Lander to a point worse to a uniform random policy. Since this is clearly discussed in the paper, I would not deem this a terrible weakness of the approach, but this might be possible to improve on.


**Weaknesses:**

Performance of SHB appears to heavily depend on the inclusion / exclusion of single environments. For example, the authors demonstrate how under SHB, the algorithm DeepQ sacrifices performance in Cliffworld and Lunar Lander to a point worse to a uniform random policy. Since this is clearly discussed in the paper, I would not deem this a terrible weakness of the approach, but this might be possible to improve on.

**Additional Feedback:**

Some of the figures in the main text are missing captions.

**Clarity:**

The paper is very well written and the high-level is easily understandable even for researchers with little RL background.

**Correctness:**

The claims, evaluation methods and experiments in this paper seem technically sound to me.

**Documentation:**

The authors provide code to reproduce the experiments in the paper. There is no additional content such as data or benchmarking code, however the proposed method is described in detail and appears to be fairly easy to use for other researchers.

**Ethics:**

There is no discussion of ethical or societal implications. Since this paper proposed a method of benchmarking rather than a new data set, it is unclear whether such a discussion is needed.

**Relation To Prior Work:**

There is only little discussion of related work, however I can not speak to how much work has been done in this area.

**Summary And Contributions:**

The paper introduces the Single Hyperparameter Benchmark (SHB), a new benchmark for comparison of RL algorithms across environments using a single hyperparameter. This benchmark shifts the focus from optimistic results based on environment overfitting to stability and generality of RL algorithms. The idea is to evaluate algorithms on a set of environments using the best hyperparameter setting across environments rather than per-environment. SHB is computationally inexpensive and easy to run.

The authors motivate and introduce SHB and evaluate it on an array of known RL algorithms. They discuss how to interpret the SHB results and how it behaves in certain settings. As an example, the SHB is used to demonstrate that Orstein-Uhlenbeck noise does not outperform Gaussian noise in the DMControl suite of tasks.

---

> ### Author Response · Authors · 2021-07-08
> **Author Response**
>
> We strongly agree that the fact that several algorithms sacrifice performance on singular domains is highly problematic. We hope that the SHB acts as a call to action as well as an empirical tool to assist in algorithmic development where such strong tradeoffs within individual domains are not necessary.
>
> We did extensively test other forms of hyperparameter selection strategy beyond maximizing average performance to understand the impact of our selection strategy on these tradeoffs (i.e. to ensure they are due to features of the algorithm and not features of the hyper selection method). For instance, in Appendix B.4 we investigate if selecting hypers to maximize performance on the worst-case environment reduced these significant tradeoffs. Unfortunately, because DeepQ and DQN both have environments where they perform poorly for nearly all hyper settings, it appears that hyper selection alone is unable to account for this sacrifice. Other experiments that were not ultimately included in the paper were: four different Condorcet voting strategies to select hypers which have some desirable tradeoff properties in social choice theory, choosing an upper-percentile over hyper settings instead of strict maximization, and choosing a lower-percentile over environments instead of strict minimization. Each of these suffered due to the underlying algorithms (DeepQ and DQN) and were unable to find appropriate hyper settings that worked well across all environments and did not exhibit catastrophic sacrifices on a few environments.

---

### Official Review · Reviewer_bH9J · 2021-07-04
**Work that is in a right direction but needs more documentation, code, explanation and clarity**

**Rating:** 5
**Confidence:** 5

**Strengths:**

The work is in the right spirit according to me. We definitely need such analysis of sensitivity to hyperparameter tuning for RL agents. It could also be used as a cheap proxy to check whether RL agents are progressing more generally to doing well over distributions of environments. So, it is broadly applicable. They also perform many experiments for the SHB vs per-environment vs subset of environments tuning comparison which is good.

**Weaknesses:**

The paper needs to make the bigger picture clearer I think. While it _is_ broadly applicable, the use-case is still not universal. The goals could be different for different people. For example, someone might be interested in dynamic tuning as in PBT (https://arxiv.org/abs/1711.09846) and only obtaining a good agent on a specific environment. While this is discussed indirectly by saying that so far research has been focused on what _can_ be done and not on generality, it's not explicitly stated. I think it needs to be made clearer that the SHB is for someone wanting to perform better on a (limited) distribution of environments.

Further, if the distribution is too large then the SHB is not cheap anymore. Someone might want good performance on a much larger set of environments than 6, e.g. 57 for Atari and then the SHB is not so cheap anymore imho. While they show experiments with DDPG on complex environments with 28 environments, they use fewer final runs (30) than the suggested number of 100 in L107. Also, I think the compute used for DMControlSuite is not mentioned in the paper. Basically, the point is that what generality is for an RL agent is not objectively defined - what number/distribution of environments at which to draw the line for SHB is a bit artificial. And if this number is too large (which it would need to be to measure "truer" general progress of RL agents), then the SHB will be slow. Even, the cited paper "Generalized domains for empirical evaluations in reinforcement learning" suggests that one may overfit to even the distribution of environments seen (page 3, column 2 in that paper) and the goal is not to be robust to an arbitrary distribution. So, the paper needs to be more explicit about its limitation that it is somewhere between true generality and per-environment specificity.

Furthermore, it needs to provide guidance on how to choose the numbers of runs per environment. Were 30 and 100 above arbitrary? How was $n_{runs}$ chosen to be 3? Was 2 tried as a value for it? Was 3 chosen because it was the lowest possible value for which the SHB worked?
With respect to choosing these numbers, the danger with complex environments environments is that they are even more variant than simpler ones and, in my opinion, would need $n_{runs}$ to be even greater than for the small environments to get proper conclusions for the SHB.

The paper also needs to provide guidance on the environment sets it is expected to be run on. Do the authors want us to only use the 6 small environments in every RL paper as a proxy measure of general RL progress? Because, as mentioned, if SHB is used on large environment sets, it could be too expensive.

**Details on hyperhyperparameters are missing**:

How were the grids of hyperparameters (HPs) selected? Why were the values on the grid for stepsize of the tile-coded agents 2^-3 larger than for the deep RL agents? Couldn't comments about cherrypicking be made at the hyperhyperparameter level for this paper? How was the subset of HPs that was studied picked? There are so many possible RL HPs. Is it because the tested HPs are more important in the authors' opinion? Could the authors provide guidance on which HPs a researcher should use when using the SHB?

Because the conclusions on individual environments were very noisy in Figure 3, I find it surprising that the mean of such noisy results was perfectly consistent. This is why I'd be interested in seeing the SHB for a different subset of hyperparameters and/or on a different set of (possibly small) environments. This would hopefully also help with answering the questions in the previous paragraph.

As an aside, the name "stepsize" really confused me because I'm used to "learning rate" and I initially thought that "hyperparameter stepsize" was the stepsize used to perform grid search over all the HPs. So, I think this needs to be clarified in the paper as well.

The repo needs a README guide on how to run experiments.

The repo could allow users to easily use the SHB by defining, e.g., a *run_SHB.py* file that takes agent(s) and environment(s) as arguments. I know that agent APIs are not standardised, but one could create an "SHB API" for agents that one should conform to and then SHB could be run out-of-the-box as long as the user's agents conform to the SHB API. This is easy to do and would aid adoption much more.

I think there should be another experiment similar to the one for DDPG that provides insight on another combination of HP and environments.

Another key aspect which I think is not explicitly touched upon is the hyperparameter tuning budget used by researchers. Given the large variance of the models used by the community today, given enough time to tune hyperparameters, any method with enough variance can, in principle solve a task as far as I understand (this is related to Section 5). So, a guideline on limiting HP tuning budget should be added I feel (Or, a metric that quantifies/normalises the performance in terms of the $n_{runs}$ and "$n_{final\\_runs}$" chosen by the researcher).



**Additional Feedback:**

The "Ethical considerations" section was supposed to be in the main paper according to the paper checklist, so this allowed the authors more than half a page extra over the limit.
The checklist item asking for "the code, data, and instructions needed to reproduce the main experimental results" is a TODO.

Here are more comments pertaining to specific line numbers from the text (I mark them as L<n> where n is the line number):

L2: "allows comparison of RL algorithms across environments using only a single hyperparameter"

Did you mean a single hyperparameter "configuration/setting"? I found it confusing that "hyperparameter" was used to mean "hyperparameter configuration" multiple times.


L16: "AIGym"

I think it's called OpenAI Gym.


L144: "For example, the parameters determining the physics of classical control domains were randomized and sampled to avoid domain overfitting (Whiteson et al., 2009),"

Reference seems wrong.


L160: "y-axis the probability"

Is it the pdf and not the probability? (Can't be probabilities, since the sum would be >1.)


L162: The 2 paragraphs beginning here seem a bit vague.
"DQN is more likely to fail than to learn a policy which solves this relatively simple environment."

What does failure mean here? Performance worse than random? Performance <100, <200, ...?

L167: "First, using the expected value of this distribution as the summary statistic does not aptly demonstrate that the poor performance of DQN on Cartpole is due to occasional catastrophic failure."

How is catastrophic failure defined?

"Instead, mean performance might lead us to wrongly conclude that DQN on Cartpole usually finds a sub-optimal policy—better than random."

How is sub-optimal policy defined? Performance <300? Sentence seems to imply it's defined as being better than random.


"An alternative might be to consider percentile statistics, or if the goal is to evaluate mean performance, to avoid drawing strong conclusions about individual runs."

I did not understand this at all. Should the sentence be split in two?

L173: "If the goal is to report mean performance, ..."

What might be the alternative goals? I don't remember seeing alternative metrics being discussed before.

L179 and 181: Grammatically wrong.
Additionally, the definitions of sensitivity and stability are not completely clear to me.


L210: "aligns empirical practice with the goals of applied RL"

These goals were not explicit to me. As mentioned before, goals might be different for different researchers.


"2) is computationally feasible even in complex environments"

As mentioned before, to me, computational feasibility seems as it would not always hold.


"3) provides novel insights on old ideas (even with small environments)"

I think this claim is correct but would prefer a bulleted list of these insights.


L223: "We use a lightly modified version of the CDF normalization method from Jordan et al. (2020),"

Could you please justify why you used a modified version?

L238: Did you mean proportional to |E|? You run theta_SHB on every environment, right?
In any case, the claim only holds if n_runs * |E| * |Theta| << n_final_runs * |E|, right?

L247: What does "added variance" mean?

L248: "This evaluation methodology provides some insight into the difficulty of tuning, though requires a sensible distribution over hyperparameters to be chosen."

Even SHB would require sensible ranges for HPs. Isn't this why the HP grid values used are different for tile-coded and deep agents?

L253: "Both of these strategies help identify algorithms that are difficult to tune, but the SHB is easier to use and computationally cheaper."

Could you please substantiate this claim with concrete numbers and what you mean by "easier"?


L273: Why was LunarLander run for 250k timesteps as an exception? Is there an objective metric to decide total timesteps? Otherwise, it feels like a per-environment tuned HP.

L275: How were the stepsizes chosen?

L282: The dataset over which the bootstrap samples were taken was unclear to me. I guess it's defined in L64 but it took me quite a while to find again.

L287: How exactly is an experiment defined? I'm not clear what is plotted in Figure 3. Is it the means of 10k values (with each of these values itself a mean over three bootstrapped values)? Or is it the mean of the final 250 runs?
The caption for figure 3 is misleading because, by itself, it seems to suggest that perfect consistency is between per-environment and across-environment rankings. Only if you know the context from the text is it clear what is meant here.

L290-292 were very confusing to me. I think this stems from saying that selecting HPs was difficult. How can this be difficult when all that needs to be done is take averages?

L297: I don't know what "counting wins" means.

L306: Could you please explicitly show the calculations? I tried to get to the figures stated in the text but couldn't.

L377: Why was the number of bootstrap samples chosen to be 1k instead of the 10k used earlier? As I understand it, this step doesn't add much overhead to the computation, whether it's 1k or 10k.

L649: "Although there is a large difference in the WalkerRun environment, we point out this may be due to the SHB trading-off performance on other environments in order to pick a single hyperparameter setting; without explicit per-environment experimentation this remains unclear."

This can be seen as a limitation of the SHB - that we no longer have clear insights on a per-environment basis. Returning to the point about different use-cases for different researchers, there's a chance that if we ever get to an "optimal" RL agent, that it will have different HPs per environment. Such an agent may be hard to test using the SHB.

L396: The claim is a bit too strong for my liking. Because, the configuration space for HPs that was swept over (defined from L696 onwards) is far from exhaustive. Also, the sigma (L700) differs from previous work, so results may not be comparable.
Further, I am curious whether the claim in L701 is based on using the SHB to decide on choosing a smaller sigma. The sentence is unclear on this and it seems some kind of human judgement might have been involved in deciding what value of sigma was a good choice. Could you please clarify?


EDIT: Updated score based on authors' changes.

**Clarity:**

The paper does need to clarify many parts of the text which are further mentioned below.

**Correctness:**

They are generally correct as I see it. But some points would need to be addressed as mentioned later.

**Documentation:**

The documentation is the part that is most lacking. The repository needs to have a README and a guide to use the code at least. Also, the classes should be better documented.

**Ethics:**

No ethical concerns as far as I can see.


**Relation To Prior Work:**

The paper discusses several important related works.

**Summary And Contributions:**

The work is aimed at standardizing the selection of hyperparameters for a distribution of environments. It does this in a simple manner that can be cheap and this appears to be a robust methodology for the subset of hyperparameters and environments evaluated. It shows how per-environment or even "subset of environments" tuning can lead to misleading conclusions on the whole set of environments one may be interested in. It also analyses the distribution of performance of RL agents on these environments.

---

> ### Author Response · Authors · 2021-07-08
> **Author Response**
>
> We thank the reviewer for the in-depth review and copy-editing notes. We have clarified the writing in several parts of the paper according to your feedback. We hope the following response helps to allay some of your concerns.
>
> We agree with the reviewer that the SHB absolutely is not an end-all empirical practice for all future empirical work in RL. There continues to be value in per-environment tuning, for instance when there is only one domain of interest (say Go, Chess, StarCraft, etc.). In our discussion in the abstract (first sentence) and introduction (second paragraph), we note that the SHB is an addition to the empirical toolkit, not a full replacement. We have modified the introduction to state this more explicitly. With regards to: “I think it needs to be made clearer that the SHB is for someone wanting to perform better on a (limited) distribution of environments”, we state this explicitly on line 23 (end of first paragraph), as well as line 79, and mathematically formulated in the paragraph starting on line 131. In all discussions about generalizing over environments, we qualify that we mean only the set of environments selected for the SHB. In Figure 5, we explicitly test the ability to generalize to environments outside of this set and conclude that none of the evaluated algorithms are capable of such generalizations.
>
> For the cost of the SHB, we note that all discussions are with respect to the alternative cost of per-environment tuning. We agree that running the SHB on 57 environments would be quite expensive, but considerably less expensive than tuning hyperparameters on 57 environments would be. The number of runs used with the SHB needs to be determined with statistical power analysis and cannot be stated ahead of time without knowing details of the environments and algorithms compared. For our experiments, 250 runs for the small control SHB was (a) sufficiently cheap to obtain and (b) useful for identifying small differences between algorithms. For the DMC-SHB experiment, 30 runs was sufficient to show with high confidence that there was no meaningful difference between OU noise and Gaussian noise.
>
> We will provide two named environment sets as examples of the SHB. We note that the final experiment of the paper uses 28 environments, not 6, as also indicated in the abstract, introduction, Section 6, and the conclusion; making clear that our goal is not to suggest that we “want us to only use the 6 small environments in every RL paper”. We use the 6 small control environments throughout Section 5 because they are computationally cheap enough to allow us to simulate thousands of applications of the SHB to evaluate the effectiveness of our evaluation methodology. We explicitly state this goal on line 257 and further describe motivations for each particular domain in Appendix C.1.
>
> Hyperparameters grids were chosen using a combination of previous literature, prior knowledge, and pilot testing to obtain approximate ranges. Statements of cherry-picking do not make sense against this paper, as we make no claims of superiority for a given algorithm. We note that the definition of tile-coding given in the Sutton and Barto textbook scales stepsizes by the number of tilings, making the stepsize range for the tile-coding agents approximately equivalent to that of the NN agents. We cannot provide guidance on which hyperparameters to tune when using the SHB, as this would require knowing the details of the algorithm where the SHB will be applied, knowledge of the investigated algorithm is up to the future experimenter and their reviewers.
>
> We have updated the README in the repo to provide more details on how to run the reproduction code, as well as to provide further documentation for the library code. We note that the primary entry script of the reproduction code is named “main.py” and provides documentation at the top of the script for expected arguments (which is also printed to console if main.py is run without arguments, an industry standard behavior); however, we have made this more explicit in the modified README.
>
> With regards to the Ethical considerations section, we searched through the call-for-papers guidelines and are unable to find (a) where such a section is required to be included at all for papers discussing best practices for benchmarks or (b) where it is stated that this must be included in the main body of the paper. Could the reviewer provide a link or quote where this is stated? We have left sufficient space in the main body of the paper to include this section.
>
> ---
>
> Edit (7/14): As we are nearing the end of the discussion period, a quick follow-up w.r.t. the claim that the paper went over the page limit. The only relevant documentation that we could find is here: https://neurips.cc/Conferences/2021/PaperInformation/PaperChecklist
>
> Which states:
> >  All supporting evidence can appear either in the **main paper or the supplemental material.**
>
> Where bolding is theirs, not ours.

---

> > ### Comment · Reviewer_bH9J · 2021-07-14
> > **Response to Paper90 Authors**
> >
> > >We have updated the README in the repo to provide more details on how to run the reproduction code, as well as to provide further documentation for the library code. We note that the primary entry script of the reproduction code is named “main.py” and provides documentation at the top of the script for expected arguments (which is also printed to console if main.py is run without arguments, an industry standard behavior); however, we have made this more explicit in the modified README.
> >
> > * I had to look quite a bit now to find this main.py. Did you mean this file: https://github.com/andnp/single-hyperparameter-benchmark/blob/main/paper/src/main.py ? I am sorry but I do not see any "documentation at the top of the script" as mentioned. There are not even docstrings in there (please see: https://realpython.com/documenting-python-code/). Could you please provide concrete links?
> >
> > * Could you please provide links or references for statements like "an industry standard behavior"? Is putting the "main.py" in the "paper" directory also an industry standard? Is a README not an industry standard?
> > Please also see https://nips.cc/Conferences/2021/PaperInformation/CodeSubmissionPolicy and https://github.com/paperswithcode/releasing-research-code for useful guides to making research code available.
> >
> >
> > >We will provide two named environment sets as examples of the SHB. We note that the final experiment of the paper uses 28 environments, not 6, as also indicated in the abstract, introduction, Section 6, and the conclusion; making clear that our goal is not to suggest that we “want us to only use the 6 small environments in every RL paper”
> >
> > Firstly, thank you for the information. Did you now update "We will provide two named environment sets" in the new version of the paper? I could not find this information.
> >
> > Secondly, you say "making clear that our goal is not to suggest". I am sorry, but in my opinion, you should not state something like that *categorically* for every person - "making clear" is a subjective thing. Your goal was *not* clear to me.
> >
> > >We note that the definition of tile-coding given in the Sutton and Barto textbook scales stepsizes by the number of tilings, making the stepsize range for the tile-coding agents approximately equivalent to that of the NN agents.
> >
> > Thank you for the information. Did you now update this in the new version of the paper? I could not find it.
> >
> > >“I think it needs to be made clearer that the SHB is for someone wanting to perform better on a (limited) distribution of environments”, we state this explicitly on line 23 (end of first paragraph), as well as line 79, and mathematically formulated in the paragraph starting on line 131. In all discussions about generalizing over environments, we qualify that we mean only the set of environments selected for the SHB.
> >
> > Thank you for pointing these out. I'm aware of those. I said that it needs to be made *clearer* than it is. It is my view and I think that that aspect of the paper can be improved.
> >
> > >The number of runs used with the SHB needs to be determined with statistical power analysis and cannot be stated ahead of time without knowing details of the environments and algorithms compared.
> >
> > Again, thank you for the information. I couldn't find this being mentioned in the new version of the paper.
> >
> >
> > >With regards to the Ethical considerations section, we searched through the call-for-papers guidelines and are unable to find (a) where such a section is required to be included at all for papers discussing best practices for benchmarks or (b) where it is stated that this must be included in the main body of the paper. Could the reviewer provide a link or quote where this is stated? We have left sufficient space in the main body of the paper to include this section.
> >
> > I must apologise for this. Thank you for your edit today providing supporting evidence in your favour. My understanding was based on the following: https://neurips.cc/Conferences/2021/PaperInformation/PaperChecklist. Question 1 (b) has a link to https://neurips.cc/public/EthicsGuidelines which says :
> >
> > >Submissions to NeurIPS are expected to include a discussion about potential negative societal impacts of the proposed research artifact or application. (For NeurIPS 2021, this corresponds to question 1c of the NeurIPS Paper Checklist).
> >
> > I assumed "Submissions" here meant the main paper.
> >
> > The added documentation actually looks decent now, even though I think the main.py doesn't seem to have proper documentation unlike what the authors mentioned. Additionally, the authors were right about the "ethical section" and they have rectified their checklist. They have also made at least some clarifications in the new version of the paper. So, overall I increase my score by 2 points.

---

> > > ### Author Response · Authors · 2021-07-14
> > > **Discussion**
> > >
> > > > I had to look quite a bit now to find this main.py.
> > >
> > > The README lays out the organization of the repo [here](https://github.com/andnp/single-hyperparameter-benchmark#organization). All code provided in `paper/` is reproduction code, which is where you will find `main.py`.  Also in this folder is a second README which has direct instructions for running the code with a single command [here](https://github.com/andnp/single-hyperparameter-benchmark/tree/main/paper#src) and further lays out the organization of that subfolder [here](https://github.com/andnp/single-hyperparameter-benchmark/tree/main/paper#organization-patterns). We want to point out that the reproduction code is _not_ intended as a library release, but rather is intended for reproducibility only (the README has been modified to make this more explicit). We would prefer to avoid modifying this code as much as possible to remain true to the code that produced the actual results in the paper (as we edit this code, the ability to reproduce goes away).
> > >
> > > The library code is well documented in the main-level README [here](https://github.com/andnp/single-hyperparameter-benchmark#cross-environment-hyperparameter-setting-benchmark). More importantly, the library code was written with readability as the primary objective with many code-level comments throughout the source with the intention of acting as a supplementary explanation for details already contained within the paper.
> > >
> > > We want to note that the paper stands alone without the code, the code is provided as a supplement to the paper (in a way similar to the appendix). We believe that the codebase itself provides additional value and contribution to the work, but our primary contribution remains to be a novel empirical procedure/benchmark and our extensive empirical results motivating this procedure.
> > >
> > > ---
> > >
> > > > Did you now update "We will provide two named environment sets" in the new version of the paper? I could not find this information.
> > >
> > > Yes, this is stated in the fourth sentence of the abstract. Additionally lines 276-277, 285-292, and 386-387 in the main body of the paper.
> > >
> > > ---
> > >
> > > > "making clear" is a subjective thing. Your goal was not clear to me.
> > >
> > > The paper explicitly runs an experiment using 28 environments (not 6). We firmly believe that the fact that our demonstration uses 28 environments leaves no doubt that we are not proposing to use 6 environments. We are unsure how to improve clarity than by demonstration. We also note, line 273 says (w.r.t. the 6 chosen environments):
> > > > We emphasize that this is not a general requirement of the [SHB] and is required only in the case of evaluating the [SHB]'s responsiveness to perturbations in the experimental process.
> > >
> > > We will further motivate in the conclusion that the SHB can (and should) be extended to further environment sets, including for instance Bellemare's "hard exploration" environment set, an environment set for off-policy policy evaluation using environments akin to Sutton et al. 2009, etc. We hope that this additional discussion helps clarify the extensibility of the SHB.
> > >
> > > ---
> > >
> > > > I said that it needs to be made clearer than it is. It is my view and I think that that aspect of the paper can be improved.
> > >
> > > As the reviewer correctly points out, clarity is subjective. It is our opinion that the explicit mathematical formulation cannot be more clear. That said, we recognize our previous response was poorly worded. We meant to say that we added _new_ qualifications throughout the paper that all conclusions can be drawn exclusively w.r.t. the tested environment pool. You can find some examples of this on lines 15, 91-92, 308, 347-350, 358, 401, 406, and the Figure 7 caption. In all of these cases, when drawing conclusions about our results we explicitly state that these conclusions are constrained to the tested environments (6 small control domains in some cases, 28 DMControl domains in others).
> > >
> > > While finding these line numbers, we identified one case where we did not sufficiently qualify our claim. We will upload a new version of the paper that corrects this (will be approximately line 419).
> > >
> > > ---
> > >
> > > For the comments about statistical power analysis and the definition of tile-coding, we point out that our paper need not be responsible for defining these concepts (they are both highly well established, TC defined in the introductory book to RL and power analysis a topic in most introductory statistics courses). We are concerned that adding this additional content makes the paper less clear through dilution. We will however add a brief discussion to the appendix, for instance where we give additional experimental details.
> > >
> > > Edit: while adding the scaling factor for TC to the appendix, we discovered it is already explicitly mentioned! Line 683. Our other mentioned changes have now been uploaded.

---

### Official Review · Reviewer_gwaH · 2021-07-05
**A well-argued proposal for an alternative way to evaluate RL algorithms**

**Rating:** 8
**Confidence:** 3
**Correctness:** I did not have any concerns about cor…

**Strengths:**

The strengths of this paper are straightforward and compelling:

[S1] The paper discusses a significant issue of practical importance to RL researchers and practitioners. Moreover, the proposed approach is simple, computationally inexpensive, and well argued for. The basic idea of using just one set of hyperparameters for an entire suite of environments is already common practice in the planning community, and RL could benefit greatly from similar community norms.

[S2] The experiments are quite thorough and addressed a number of objections that I had while reading the methods section (e.g. Does it matter if we aggregate scores across environments using something other than a summation during tuning? Does using a single hyperparameter setting actually change the rankings of algorithms? etc.).

**Weaknesses:**

However, I still objected to one of the major arguments in the paper and felt there were some lesser weaknesses in other areas:

[W1] The paper argues in various places that SHB does not require as much hyperparameter tuning as existing benchmarks, and in particular trumpets the fact that the experiments in Section 5 obtain good results using only three tuning runs. The conclusion takes this further, arguing that it allows "small university labs and tech giants alike to conduct rigorous and meaningful comparisons". My interpretation of these claims (which may not be the intended one) is that SHB-type approaches that average over many environments are not as sensitive to hyperparameter tuning as existing approaches, and so confer less benefit to well-resourced labs that currently do absurd amounts of tuning to make their algorithms shine.

Insofar as this is the intended argument, I am not yet convinced by it. In my experience, benchmarks are used throughout the development cycle to make design decisions and choose default values for hyperparameters (which may or may not be tuned over afterwards). There's a risk that an algorithms researcher using SHB might accidentally (or intentionally) do a huge amount of manual tuning on the SHB score during algorithm development by using it to make design decisions (e.g. should I use a replay buffer or not? What should be the base architecture for my network? etc.). The resulting ranking might be stable across tuning runs, but would give the new researcher's algorithm an unfair advantage over baselines that may have originally been developed to target a different population of environments. Figure 6 partially addresses this concern, but I'm not convinced that the limited number of hyperparameters considered (LR, replay buffer size, target network refresh rate) really capture the number of degrees of freedom that are typically available in deep RL. SHB definitely has value for an "honest" evaluator comparing existing algorithms on "held-out" combinations of environments, but I'm concerned that the paper overstates the degree to which it will address the current compute arms-race among benchmark-minded researchers.

[W2] This is not so much a weakness as an opportunity, but I feel that there are a few places where this paper could do more to encourage comparability of results across papers. One is in the choice of environments: as far as I can tell, the paper is trying to be somewhat agnostic to precisely what environments go into the SHB population, so that researchers could apply this methodology to any other benchmark suite. I think there is value in defining particular combinations of benchmarks and giving them "official" names (e.g. "classic control SHB" for the environment population in section 5, or "dm_control SHB" for the environment population in section 6). This would encourage researchers to use those specific combinations in future papers, rather than dropping environments that have inconvenient results.

I think there would also be some value in creating a predefined $\mathcal P_E$ for each environment in the paper so that future papers can copy those results instead of re-evaluating every algorithm anew each time. This might create some perverse incentives (e.g. to dedicate tuning budget to environments with weak baselines), but could be worth it if it encourages researchers to compare to more algorithms, since they don't have to re-run each one. It would also serve as another incentive for them to use the same combination of environments as previous papers instead of picking combinations that are favourable to them.

[W3] There are some lesser clarity issues that I discuss below.

**Additional Feedback:**

 On the whole I enjoyed reading this paper and think that it would be a valuable addition to the track.

------

(review last checked/updated 2021-07-19)

**Clarity:**

[CL1] I was confused by the terminology in the title and introduction. Usually "hyperparameter" refers to a particular control knob for an algorithm such as, e.g., the step size of SGD. I initially assumed "single hyperparameter" meant that the algorithm was only meant to expose a single control knob, but that this knob could be tuned separately for each environment just like the (many) control knobs of existing algorithms. This misunderstanding was clarified fairly quickly, but using "hyperparameter setting" instead of "hyperparameter" might help readers who have the same confusion as me.

Similarly, I assumed that the "Single Hyperparameter Benchmark" referred to a particular set of environments that the paper was proposing (which is what I usually think of as a "benchmark"), but then was surprised in Section 6 when it was used to a different set of environments. I would personally find "single setting of hyperparameters" easier to understand than "single hyperparameter", although I admit "SSoHB" does not roll off the tongue.

[CL2] I didn't understand how the normalization was defined on lines 222 to 233 during my first read-through. After reading Jordan et al. & coming back to this I think all the necessary information is in fact present in the paragraph, but it took me a while to parse. Explicitly stating that $N(G) = CDF(G,E)$ might help.

**Documentation:**

Github documentation could benefit from some more explanation in the README, as well as some more examples (e.g. of how to register custom environments). Docstrings would also help.

**Relation To Prior Work:**

I'm not aware of missing related work, and the existing comparisons seem adequate.

**Summary And Contributions:**

This paper proposes a new methodology for evaluating RL algorithms, which it terms the Single Hyperparameter Benchmark (SHB). Rather than separately tuning each algorithm on each benchmark environment, SHB advocates doing a small amount of tuning for each algorithm to find a single hyperparameter setting that gets high average return across all environments. Experiments show that this approach produces stable rankings across executions of the tuning process, even when using just a few seeds for tuning.

---

> ### Author Response · Authors · 2021-07-08
> **Author Response**
>
> We want to thank the reviewer for their detailed and insightful review. Following the reviewer’s suggestions, we have elected to change the name of the benchmark to make more clear the distinction between hyperparameters and hyperparameter settings. For consistency, in our responses we will continue to use the abbreviation SHB, but the future iteration of the paper will use the abbreviation CHS (Cross-environment Hyperparameter Setting benchmark).
>
> ---
>
> [W1] One of our goals is to reduce the extent that exhaustive tuning (brought to you by big labs) can influence results. We agree completely with the reviewer that the SHB does not eliminate this influence, however. To understand the extent that exhaustive tuning influences conventional methods versus the SHB, and to motivate that the SHB lessens this advantage, we designed the experiments in Section 5 such that the tile-coding agents receive an unfair amount of tuning compared to the NN agents. Specifically, the TC agents receive 50% more opportunities to select ideal hyperparameter settings. In Figure 4, we show the difference between conventional reporting methods and the SHB, illustrating that the advantage afforded to the more highly-tuned algorithms is larger for conventional methods than the SHB on average, driven largely by extreme overfitting on Cartpole and Mountain Car.
>
> To address this limitation, we have extended our discussion in the Ethical Considerations section (Appendix A) to point out advantages afforded to labs with access to greater resources and that the SHB does not eliminate these advantages. We also clarify our statement in the conclusion that the SHB helps small university labs by providing a more challenging benchmark using small and inexpensive domains, but not necessarily by levelling the playing field for tuning opportunities (e.g. small control domains are often thought too simple to provide meaningful insights, however in using the SHB, these domains become considerably more challenging and provide a new avenue for meaningful development without scaling up the required compute).
>
> [W2] This is an excellent point. In Appendix C.1 we motivate our reasoning behind the inclusion of each small control domain into the SHB. We have given this benchmark a name (Small Control CHS) and now use this name throughout the paper, referencing Appendix C.1 for the motivation behind the choices in developing the SC-CHS. Likewise for the DMC-CHS (Deepmind Control CHS). We additionally will add these explicit environment suites to the code library to further cement their inclusion. We will add details of the selected \theta_\text{shb} for each algorithm on both benchmarks to further assist replicability of the results without the need for extensive hyperparameter tuning in follow-up works. Finally, we will add the data necessary for computing $\mathcal{P}_E$ to the code-release for both benchmarks.
>
> ---
>
> [CL1] We have clarified the distinction between hyperparameter and hyperparameter setting in early critical sections of the paper, including changing the name of the benchmark itself to the Cross-environment Hyperparameter Selection benchmark (CHS). Likewise, we have clarified in the introduction that the SHB (now CHS) is environment agnostic, but that we introduce two concrete instantiations of the CHS in this paper and encourage development of future instantiations (for instance for comparing exploration algorithms).
>
> [CL2] We really appreciate the thoroughness of the reviewer here! We have reworked this description of the CDF scaling to be a bit more clear, taking advantage of the additional page of allowed space.

---

> > ### Comment · Reviewer_gwaH · 2021-07-20
> > **Changes**
> >
> > Thank you for making those changes. I see your point about how Figure 4/Section 5 relate to [W1], although I don't think they provide strong empirical evidence for the insensitivity of SHB to tuning. In a sense, the curve for "impact on SHB of each level of tuning" is only being evaluated at two points (48 vs. 72 tuning runs), and the results are confounded by the use of different types of function approximation (NNs vs. tile coding). To firmly establish the claim about hyperparameter sensitivity, I think it would be necessary to have an experiment that sweeps different levels of tuning in a uniform way across all algorithms. Other than that, I think this is good work, and have updated my score to be more in favour of acceptance.

---

### Official Review · Reviewer_fNuV · 2021-07-05

**Rating:** 6
**Confidence:** 3
**Correctness:** Yes
**Clarity:** Yes

**Strengths:**

The proposed evaluation protocol makes the evaluation of RL algorithms more robust w.r.t. statistical errors and also focus more on generalization rather than overfitting to individual environments. SHB also makes RL evaluations less compute-intensive via only sweeping hyperparameters for 3 runs. SHB is very important for RL community since RL algorithms have been known for sensitivity to random seeds and susceptible to statistical errors. I think it would be of great significance for the whole RL community.

**Weaknesses:**

The authors claim SHB as a benchmark but it seems it is rather just an evaluation protocol. It would be best to clarify it in the paper. Moreover, evaluations of RL algorithms on continuous control tasks such as locomotion and robotic manipulation are missing, which makes the evaluation less thorough. I strongly suggest that the authors should include these evaluations and also add results of more recent RL algorithms such as TD3, SAC, etc.

**Additional Feedback:**

See above

**Documentation:**

Yes

**Relation To Prior Work:**

Yes

**Summary And Contributions:**

This paper proposes a new benchmark for RL that uses one set of hyperparameters across all environments. The authors argue that tuning hyperparameters per environment requires a large number of runs in order to get rid of the statistical errors, which also makes running RL algorithms quite compute-intensive. Therefore, the authors propose to tune a single hyperparameter across environments with a preliminary set of runs for hyperparameter selection and then evaluate the selected hyperparameter for hundreds of trials on each environment for the final performance. The authors perform the proposed evaluation protocol SHB on classical control tasks and RL algorithms and find that the algorithm rankings found by SHB are consistent across all environments. The paper also finds that the change in performance for each algorithm on every environment when using the SHB
versus conventional per-environment tuning is quite large on each environment, suggesting that RL algorithms are not quite able to be generally applied to all environments.

---

> ### Author Response · Authors · 2021-07-08
> **Author Response**
>
> We thank the reviewer for their feedback! We agree that the SHB is a more generic benchmark, which emits specific benchmarks for any choice of environment pool or evaluation statistic. In the paper, we discuss two such specific benchmarks (one with classic control problems, the other with the full DM Control suite) which both fit classic definitions of a benchmark. We will make the wording of these two benchmarks more explicit, for instance by naming them.
>
> We want to clarify that the goal of Section 6 is not to provide a benchmark of recent algorithmic developments across many domains. Rather, the goal is to answer a hypothesis that has already been set up in the literature and thereby demonstrate the utility of the SHB. The DDPG, D4PG, and TD3 papers all use Mujoco-based domains to evaluate their respective claims about the utility of OU noise, so to evaluate their hypothesis on the DM Control suite is sufficient. We strongly agree that expanding the scope of the hypothesis (e.g. by including more algorithmic developments, or by looking at domains with different attributes) would be a valuable contribution to the field of continuous control, but is outside the scope of a paper introducing a new empirical methodology.

---

### Decision · Program_Chairs · 2021-07-27

**Decision:**

Reject

**Comment:**

This paper created a lot of discussions, involving the reviewers and multiple area chairs.
The paper received quite high scores by several reviewers, with 3 non-confindent (confidence:3) votes for acceptance (scores:6,8,8) and one very confident (confidence:5) vote for rejection (score:5).
I therefore read the paper myself in detail, discussed with the reviewers, and also with another area chair.

In these discussions, there was consensus that the paper points into the right direction, but that some points are problematic and could have undesired consequences rather than facilitate better experimentation in the RL community. There are no conditional accepts in this track, but I would expect that the paper will get accepted in round 2 of this track if it addresses the remarks below.

The paper's main contribution is a new empirical protocol. The authors clearly have a good point in suggesting to move beyond empirical protocols that perform unbounded hyperparameter tuning separately for every environment. However, the proposed protocol also has some issues (please see below for details), which is problematic since many reviewers would probably ask authors to follow it. (Note that this NeurIPS benchmarking track is likely going to be the most credible resource for benchmarking in machine learning, so the protocol would be accordingly prominent.)

In detail, the protocol / paper has the following issues:

1. The paper's suggested budgets are very high: 30 seeds for every hyperparameter setting and every environment in the preliminary phase. With a single run sometimes taking several days on complex environments this policy would (a) contribute substantially to the heavy carbon emissions bill of AI research and (b) disproportionally disadvantage researchers in typical academic institutions that do not have thousands of machines for this tuning.

2. Likewise, the suggested budget for the final evaluation is extremely high. Not even DeepMind, despite its access to large compute, ran 100 seeds for all of their Atari environments.

Remark: The authors might feel that 1. and 2. are unfair since the numbers 30 and 100 are never explicitly mentioned as suggestions in the paper, but since they are used without a discussion that these numbers need to be adapted to the case at hand, future reviewers would likely ask for exactly these numbers of runs for the papers they are reviewing. (If they asked for very cheap experiments like the ones the authors are running throughout most of the paper that would of course be fine, but reviewers tend to ask for more expensive experiments, such as Mujoco, Atari, ProcGen, etc, and it should be clear from the paper that these are not the domains for which 100 repeats are suggested.)

3. The suggested protocol is wasteful as it does not take into account modern hyperparameter optimization techniques: it speaks about "parameter sweeps" throughout, i.e., grid searches. There are far more computationally efficient hyperparameter optimization methods, such as Bayesian optimization, multi-fidelity optimization, etc. Baking the least sample-efficient hyperparameter optimization technique (grid search) into the experimental protocol is not a good idea.

4. The paper misses the opportunity to make the connection to the rich literature on algorithm configuration that has studied the problem of effectively finding the best hyperparameter setting across tasks for over a decade. See, e.g., the ICML 2019 tutorial on algorithm configuration, for an overview. Algorithm configuration is basically an extension of hyperparameter optimization that tries to find a single hyperparameter setting which works well across tasks -- precisely the setting the authors propose.
4a. Algorithm configuration methods automatically allocate more budget to better-performing hyperparameter settings, evaluating poor hyperparameter settings only on a few tasks and thereby substantially reducing the high cost of the authors' proposed method.
4b. Algorithm configuration methods also use the concept of "adaptive capping": terminating poorly performing runs early to save even more compute.

Remark: this relationship to algorithm configuration does not mean that the authors' experiments need to use the latest algorithm configuration methods, but it would be good to make the connection to make the point that the proposed experimental protocol can indeed be computationally inexpensive.

Relatedly, the argument the authors make about the computational cost of the standard protocol and the proposed protocol appears broken: in standard tuning, to avoid overfitting one should also do less runs for each of the hyperparameter settings than for the final evaluation, so the computational complexity of the two protocols is the same. (With algorithm configuration, particularly 4a, actually the authors' protocol would be faster again, since not all tasks need to be evaluated for poor hyperparameter settings.)

(continued below)